# Sliding Shear Failure of Basement-Clamped Reinforced Concrete Shear Walls

**DOI:** 10.3390/ma17164111

**Published:** 2024-08-20

**Authors:** Harald Schuler

**Affiliations:** Civil Engineering Department, FHNW University of Applied Sciences and Arts Northwestern Switzerland, Hofackerstrasse 30, 4132 Muttenz, Switzerland; harald.schuler@fhnw.ch

**Keywords:** earthquake analysis, RC shear wall, basement wall, sliding shear failure, flexure–sliding interaction

## Abstract

After an earthquake struck Chile in 1985, sliding shear failure was observed in the clamping zone of stabilising walls. Even houses with only four storeys can fail due to sliding shear below basement ceilings. This type of failure has received little attention in the past; however, it leads to premature failure of the clamping effect in the basement box. Bending deformations cannot fully develop, which significantly reduces the seismic safety; it is thus important to learn more about the sliding shear failure of basement-clamped shear walls. The overall behaviour is analysed based on three large-scale shear wall tests. The deformation states around the sliding shear zone are evaluated and a simple estimate is given of when sliding shear failure can occur.

## 1. Introduction

For buildings of medium height (three to eight storeys), shear walls are often used to stabilise against earthquakes. Often, these walls are clamped into the rigid basement box to transfer the horizontal loads into the basement and afterwards into the underground. The walls consist of two parts, a cantilevered part and a clamped part. Often, the cantilevered part is rather slender, and the clamped part is rather squat. Several experiments on more slender cantilever shear walls are available in the literature [1,2,3,4,5,6,7,8,9,10]. Other experiments focus on squat cantilever walls [11,12,13,14,15]: Synge [12], Pauley et al. [13] and Luna et al. [14], where Synge investigated a wall with an aspect ratio of h_w_/l_w_ = 0.57 (Wall 1) and Luna et al. a wall with 0.54 (SW 8). Both walls failed under sliding shear. Salonikios et al. [15] investigated walls (LSW 2 and LSW 3) with an aspect ratio of h_w_/l_w_ = 1. Their walls did not slide and failed due to flexure. However, the wall with an additional axial force (LSW 3) lies in the vicinity of a sliding shear failure, as Schuler and Trost [16,17] showed. They analysed the effect of the amounts of concrete and steel on the siding shear resistance with a mechanically-based approach. Schuler and Trost postulated that most of the resistance (over 80% of their studied walls) occurs in the compression zone. Sliding shear resistance, together with the flexural resistance, is plotted over the curvature in the sliding zone, which makes the interaction between sliding shear and flexure visible.

Shear walls clamped into the basement are not found in the literature. A test set-up was developed to analyse such walls, which consist of both a cantilevered part and a clamped part. In a separate publication [18], the walls are analysed due to flexure and interaction with shear (tension shift effect). Figure 1, on the right, shows—schematically—an example building. The basement wall must transfer the bending moment within a short distance into the basement box, resulting in a high shear force in the basement wall. At the construction joint below the basement ceiling, both the shear force and the bending moment are high. From flexure, a crack opens in the range where the steel yields and afterwards, under reverse loading, the crack closes again; however, not completely. This reduces the aggregate interlock resistance of the concrete decisively, which makes up the main part of the sliding shear resistance [16,17]. The steel’s dowel action resistance in the tension zone plays only a minor role. The steel is already in the plastic range due to flexure and has no further resistance capacity. In three large-scale tests with a scale of 1:1.5 compared to the real size, the interaction of the cantilevered and clamping wall sections is examined. In particular, the sliding shear behaviour in the construction joint between the basement wall and the basement ceiling is analysed. Figure 1 (middle and right) shows the test specimen schematically, the deformations and the cutting forces. As the deformed specimen in the middle shows, the sliding shear displacement moves in the same direction as the top displacement. This is according to the theory and was also seen in the experiments.

This publication aims to analyse a possible sliding shear failure of walls consisting of a cantilevered part and clamped part. The following questions are of interest:-Where are flexural cracks localised and how do they initiate sliding shear failure? How advanced is the deformation and the plastic strain around the construction joint before sliding shear displacements occur?-How is the friction resistance in the compression zone of the construction joint? How is the friction resistance in a zone in which the cracks do not close completely when the load is reversed?

## 2. Experimental Set-Up and Test Specimens

### 2.1. Experimental Set-Up

The test facility is shown in Figure 2. The figure shows the shear wall, the strong floor, the reaction walls below and above the strong floor and the clamping equipment on the force, middle and bottom axis. Forces are applied via the hydraulic cylinder. On the middle and bottom axis, the wall is clamped and connected to the horizontal supports. The clamping is achieved via prestressed rods. Five of six horizontal supports on the south and north sides of the wall are freely movable in the vertical direction. On the north side of the middle axis, the vertical support is located, achieved by a steel hinge. Thus, each axis can freely rotate without any constraint for a free deformation behaviour of the walls.

Figure 3 (top left) shows the front side of NW 1 in the test facility. On the front side, the optical measurement reflectors are glued. One can also see the designated south side (S) and north side (N) of the walls. In Figure 3 (top right), the back side of the specimen’s cantilever part is shown. The aluminium frame, screwed to the ceiling cutout, is used to fix the six displacement transducers on the cantilever part of the wall. In the clamped part, the displacements are measured in the same way. Thus, all the displacements can be recorded in relation to the ceiling cutout. Rigid body rotations are automatically eliminated.

### 2.2. Shear Walls

The investigation comprises three clamped shear walls; the reinforcement ratio and aspect ratio are varied. The walls are sufficiently designed against shear forces to focus on the interaction between bending and sliding shear. They consist of cantilevered and clamped parts, between which the ceiling cutout is located. The thickness of the cutout is 500 mm, and the height is 400 mm. Vertical reinforcement is continuous over the total height without lap splicing. The idea is to focus on pure sliding shear failure and avoid the influence of additional effects. An overview of the geometry and reinforcement ratios of all the walls can be found in Table 1. The walls are specified subsequently. The formwork and reinforcement plans are given in Figure 4.

NW 1

NW 1 represents a wall of an existing building. It contains a small amount of longitudinal reinforcement with a reinforcement ratio of 0.44% and no boundary reinforcing. The aspect ratio of the clamped part is h_w_/l_w_ = 1.03 and of the cantilevered part is 2.04.

NW 2

NW 2 is designed to represent a ductile earthquake wall of a new building. It contains boundary reinforcing with a reinforcement ratio of 2.23% over a length of 0.15 times the length of the wall (270 mm). In the web, the reinforcement ratio is approximately the same as for NW 1. In the boundary zones, additional stirrups are inserted to confine the compression zone to increase the ductility. The aspect ratios are equal to NW 1, about one for the clamped part and two for the cantilevered part.

NW 3

NW 3 has about the same longitudinal reinforcement ratio as NW 1. The difference to NW 1 lies in the length; NW 3 has a length of 2.4 m, while the length of NW 1 is 1.8 m. Thus, the aspect ratio of NW 3 is smaller, with h_w_/l_w_ = 0.78 for the clamped part and 1.53 for the cantilevered part.

In Table 1, the geometry and reinforcement ratios are summarized for all the walls. The height h_base_ is defined as the distance between the support axis in the middle and at the bottom. The height h_up_ is the distance between the upper load axis and the middle support axis. The axes are shown in Figure 2. Table 2 lists the material properties of the concrete; Table 3 lists the material properties of the steel. The walls were fabricated in four stages along the vertical axis. The concrete properties are given for Section 1 and Section 3, which correspond to the concrete in the basement wall and the cantilever wall part adjacent to the ceiling cutout. The mean value derives from at least five material tests. The compression strength of the concrete was measured on cubes with an edge length of 150 mm. As a basis of the concrete tensile strength, the splitting tensile strength was measured on cylinders with a length of 300 mm and a diameter of 150 mm. The material properties of the reinforcement bars are obtained from the bilinear approximations of the stress–strain relationships measured in the tension tests.

Figure 5 shows the construction joint between the basement wall and the basement ceiling. The concrete was compacted with a vibrating bottle and not further roughened, as is often performed in practice. The largest grain size of the concrete is 16 mm. As the picture shows, the roughness is not pronounced. The height differences across the joint are in the range of up to 4 mm.

### 2.3. Measurement Technique

The applied measurement equipment consists of 12 inductive displacement transducers and an optical measurement grid with a horizontal and vertical distance of 200 mm (see Figure 6).

#### 2.3.1. Measurement with Inductive Displacement Transducers (Back Side)

Six displacement transducers were mounted on a frame above the ceiling cutout; another six were mounted on a frame below the ceiling cutout. Hence, the measured displacements are with respect to the ceiling cutout. This allows measurement of the displacements in the cantilevered and clamped parts separately from each other. Horizontal displacements are measured on the load axis, on the lower support axis (green dots in Figure 6 left) and, for the sliding shear displacements, 200 mm above and below the ceiling plate (blue dots in Figure 6 left). The vertical displacements on the north and south halves of the wall are measured at the same heights as the horizontal displacements. With the displacement transducers near the construction joints (red dots in Figure 6 left), the relative Y-displacements are measured; these are then extrapolated for the values at the edges.

#### 2.3.2. Optical Measurement (Front Side)

With eight cameras, the displacement field is measured over the total wall. Thus, reflectors are glued on the wall with horizontal and vertical distances of 200 mm between them. Figure 4 right shows the grid. The reflector points on the ceiling cutout could not be used because they were partially obscured by the steel structure. For the middle axis position, displacement measurement of the displacement transducers was used. The system can measure displacements in the X-, Y-, and Z-directions; for the displacement field, the X- and Y-directions (vertical plane) are needed. In this publication, the vertical displacements around the ceiling cutout are analysed along the red dashed lines in Figure 6.

### 2.4. Loading Protocol

The loading protocols were set up according to the known load history from [19]. A displacement control was chosen based on the actuator displacement in order to take into account both displacements, from the cantilevered and clamped parts of the wall, and to have a safe control system. This may result in certain deviations from a measurement based purely on the wall displacements. The loading protocol for NW 1 is given in Figure 7. The load stages are: cycle 1 = 0.75F_y_; cycle 2 = δ_act_(0.75F_y_)·4/3·2; cycle 3 = δ_act_(0.75F_y_)·4/3·3. δ_act_ is the actuator displacement. For each load stage, two cycles are driven, e.g., in stage 1, cycle 1-1 and cycle 1-2. The same load steps were applied to NW 2. In the case of NW 3, only two load stages were run since failure had already occurred in the second load stage. In the first cycle of each load step (cycle X-1), the actuator was stopped, cracks were signed and pictures were taken at the maximum north and south deflection. The interruption of the load application is represented by a horizontal line in the loading protocol.

## 3. Results

The results of the investigation of the three shear walls are presented below. The load application is defined by the load protocols. In the case of displacement control (cycles 2 and 3), the cylinder displacement is used. The clamping stiffness is not the same in the north and south directions. For a horizontal force in the south direction, the prestressing bars stretch so that the support points are softer than for a horizontal force in the north direction. There, the force can be applied directly to the steel structure. As a result, the forces in the south direction are smaller than in the north direction for the same deflection of the hydraulic cylinder. Thus, a completely symmetrical load application could not be achieved. For the author, it was more important that the clamping could be applied in the basement without additional vertical constraints.

### 3.1. NW 1

#### 3.1.1. Load–Drift and Sliding Shear Behaviour

Figure 8 shows the load–drift behaviour of the cantilever wall (left) and clamping wall parts (right). On the top, measurements in the upper load axis (3480 mm above the ceiling cutout in Figure 6) and lower support axis (1660 mm below the ceiling cutout in Figure 6) are shown. In the middle, measurements adjacent to the ceiling cutout are given. As shown in Figure 6, the displacement measurements around the ceiling cutout are located both 200 mm above and below. These displacements approximately correspond to the sliding shear displacements. The bending component within this height is very small and is therefore neglected. The load–drift behaviour from the sliding shear is given in the middle of Figure 8. No sliding shear displacements occur in NW 1, thus the differences between the total and the sliding shear displacements (bottom of Figure 8) are almost the total displacements (top of Figure 8); only displacements from flexure and tension shift occur. Figure 9 top and middle plots the vertical (relative Y-displacements) over the horizontal (sliding shear) displacements above and below the ceiling cut. The displacements are measured over a length of 200 mm. The vertical displacement is referred to as a relative Y-displacement and is abbreviated as Y-displacement. The Y-displacements above and below the ceiling reach up to 4 mm. The maximum Y-displacement occurs in the cantilevered part of the wall (see Figure 9 top to middle). In load level 3, the Y-displacement in the clamped part is somewhat obstructed because the wall is resting on the bottom. Figure 9 bottom shows the permanent Y-displacement (after unloading) in the wall centre. In load cycle 2-1 and 2-2, the permanent Y-displacement is a maximum of 1.6 mm, where the roughness of the construction joint is up to 4 mm. Finally, the sliding shear displacements are not greater than 1.2 mm, which means that no sliding shear failure occurs. The wall fails by flexure. The shear force is too small to initiate a sliding shear failure.

#### 3.1.2. Cracking and Longitudinal Deformations around the Construction Joints

In Figure 10 top, the crack pattern of NW 1 is given, showing horizontal cracking to the edge, where flexure dominates. In the centre of the wall, or towards the neutral axis, the cracks run inclined. There, the shear stress increases, while the flexural stress decreases. Figure 10 bottom shows vertical deformations in different horizontal sections. The Y-displacements on the tension side around the construction joints (Y-displacements at 0.4 m and −0.4 m) are in cycle 2-1-N nearly 2 mm, 100 mm away from the edge (Figure 10 (bottom left)). In the following cycle 2-1-S, the Y-displacements increase to values of up to 3 mm and the closures on the compression side are becoming smaller (Figure 10 (bottom right). Plastic Y-displacements remain in the construction joint. In adjacent sections, the Y-displacements are significantly smaller (0.6 m and −0.6 m, etc.).

### 3.2. NW 2

#### 3.2.1. Load–Drift Behaviour, Sliding Shear and Longitudinal Displacements

Figure 11 shows the load–drift relationships. The picture in the middle right shows the drift from sliding shear in the clamped wall section. First, sliding shear displacements occur already in cycle 2-1 with the load to the south. After the construction joint in the south edge of the wall is opened under a load to the north, the first sliding shear drift occurs when the wall is loaded in the reverse direction. The crack in the south does not close completely and the basement wall slides to the south at the construction joint. In the following cycles, the joint opens successively and the sliding shear displacement becomes more and more pronounced. This is addressed Section 3.2.2. In cycle 3-2, the drift from sliding shear develops up to a drift of 0.7% and the strength drops down to approx. 80% of the maximum value. Figure 11 top shows that the main flexural + tension shift displacements already occur in cycle 2-1. The following cycles, which are shown in colour, show no pronounced increase in the flexural + tension shift displacements. The wall begins to slide and loses its clamping load-bearing capacity in the basement; horizontal stabilisation of the building is no longer possible with the wall.

Figure 12 shows the curves of the relative Y-displacements versus sliding shear displacements. The relative Y-displacement is here defined as the longitudinal displacement measured over a measurement length of 200 mm (see also red dots in Figure 6). The relative Y-displacement is given at the centre of the joints above and below the ceiling cutout (top left and middle left). One can see that a successive increase in the centre of the wall occurs, which means that the crack in the joint also opens. The top and middle diagram on the right side show the relative Y-displacements at the north and south edges of the wall. If the north edge is under compression, the cracks does not close completely. A lengthening of between 2 and 4 mm remains after cycle 2-1 (left side of diagram top right). If the south side is under compression, the lengthening is smaller (right side of diagram middle right). In the diagram on the bottom right, the remaining lengthenings are in the compression zone, resp. the crack closures are given. Especially for the loading to the north, clear permanent lengthenings are visible. Figure 12 (bottom left) shows the permanent relative Y-displacements in the centre of the wall (after unloading). Especially in cycles 3-1 and 3-2, great lengthenings occur in the clamped part of the wall. This is also where the sliding shear displacements increase. More details about the deformations in a crack for the opening and closure part can be found in [17]. The closure of cracks is prevented by the aggregates, which cannot go back into the gaps from which they came when they slide at the same time. The aggregate interlock resistance is reduced.

#### 3.2.2. Crack Pattern and Longitudinal Deformations

Figure 13 shows the crack patterns. Compared to NW 1, more cracks occur, more cracks are inclined and, in the boundary element, secondary cracking occurs. The load capacity is more than twice that of NW 1, leading to this difference. A detailed analysis of these observations is performed in a separate publication [18]. In the publication here, the construction joint above and below the ceiling cutout is of interest to analyse sliding shear failure. Figure 13 (middle left and bottom left) shows the Y-displacements in cycle 2-1 for loading to the north. The lengthening on the tension side (x = −0.8 m) is ~2 mm in the clamped part, smaller than in the cantilevered part with ~3 mm. In this half cycle, no sliding shear displacements occur. In the following half cycle to the south—Figure 13 (middle right and bottom right)—the lengthening in the tension zone (x = 0.8 m) is substantially greater: ~6 mm in the clamped part and ~2.5 mm in the cantilevered part. The compressing of it in the compression zone (x = −0.8 m in Figure 13 bottom right) is a little smaller than in the previous cycle (x = 0.8 m in Figure 13 bottom left). This is one reason why sliding shear displacements start at this load stage (cp. Figure 11 (middle right)). One can also see that the Y-displacements in the sections where Y = −0.4 m are significantly greater than for the adjacent sections, Y = −0.6 m and −0.8 m. This further favours sliding shear failure in the construction joint.

The main sliding shear resistance is built up via the aggregate interlock in the compression zone (cp. [16]). A permanent crack opening reduces this aggregate interlock resistance and hence the sliding shear resistance.

#### 3.2.3. Photos of the Sliding Shear Zone

Figure 14 shows the cracking around the construction joint between the ceiling cutout and the basement for different cycles; the relative Y-displacement (longitudinal displacement) is measured over a distance of 200 mm. This means that there is crack opening of the construction joint, and additional crack openings in the 200 mm wall segment are included. However, the main opening occurs in the construction joint. The picture on the top shows the cycle 2-1 to the south, where the first sliding shear displacement was recognized with about 2.5 mm. This is the cycle after the first plastic lengthening of the tension reinforcement (2-1-N).

In cycle 3-1-N (cycle 3-1-S was driven before this cycle), the sliding shear displacement was approximately 10 mm. The relative Y-displacement (over a distance of 200 mm) in the centre of the wall was about 5 mm. It is estimated that at least half of this localises in the construction joint, which would be a crack opening of about 2.5 mm. One also can see that spalling of the concrete cover starts due to transverse tension. The picture on the bottom shows the joint after cycle 4, a further cycle that is not included in the evaluation before. It shows the final damage state. Through forward and backward sliding shear displacements, the concrete spalls and rubs off along the joint. The sliding shear resistance is reduced from cycle to cycle.

### 3.3. NW 3

#### 3.3.1. Load–Drift Behaviour, Sliding Shear and Longitudinal Displacements

Figure 15 shows the load–drift relationship of NW 3. The wall has the same reinforcement ratio as NW 1 but is 1.33 times longer. The diagrams on the top show the flexural (incl. tension shift) drifts. For loading to the north, the cantilever part deforms mainly (left diagram); for loading to the south, the clamped part (right diagram). The reason for this is unsymmetric cracking. The diagrams in the middle show the drift from sliding shear, left in the cantilevered part and right in the clamped part. The maximum sliding shear drift occurs in the clamped part in cycle 2-2, with 0.43% for a deflection to the north. This corresponds to a siding displacement of eight millimetres. In this load stage, the sliding shear displacements increase significantly while the flexural displacements already decrease. The load drops to about 70% of the maximum value. The drifts from the total displacements are given on the bottom.

Figure 16 shows the longitudinal sliding shear displacement curves in the cantilevered part (top) and clamped part (middle). Again, the first sliding shear displacements occur in cycle 2-1 to the south, after the first plastic lengthening of the tensile zone in the cycle before. Figure 16 (bottom) shows the permanent lengthening after the cycles. After cycle 2-1, the plastic lengthening in the centre is not pronounced, but after cycle 2-2, the great sliding shear displacements occurred.

#### 3.3.2. Cracking and Longitudinal Deformations around the Construction Joints

Figure 17 (top) shows the crack pattern and Figure 17 (bottom) the longitudinal deformations of NW 3. The cracks are distributed asymmetrically: for a deflection to the north, cracks develop above the ceiling cutout in the south (see Figure 17 top left). Only a few cracks appear below the basement ceiling in the south. For a deflection to the south, cracking occurs below the ceiling cutout on the north side (see Figure 17 top right). Above the ceiling, only a few cracks emerge. This asymmetrical behaviour is due to the position of the vertical support, which is located on the central axis at the northern edge of the wall. The dead load of the hydraulic cylinder and the horizontal stilt in the lower bearing axis also play a role. This creates additional pressure above the ceiling cutout and additional tension below the ceiling cutout. This leads to the asymmetrical crack distribution. However, this vertical support was the simplest solution to create a constraint-free bearing.

Nearly equal lengthenings of about 4 mm occurred above and below the ceiling cutout in cycle 2-1-N (X = −1.1 m section and Y = 0.4 m/Y = −0.4 m in Figure 17 bottom left). For the subsequent cycle to the south, 2-1-S, a large single crack of about 16 mm occurred in the clamping part (X = 1.1 m section and Y = −0.4 m in Figure 17 bottom right). As for NW 2, the cracks did not close completely in the compression zone. Sliding shear failure occurred due to the reduced aggregate interlock resistance.

#### 3.3.3. Photos of the Sliding Shear Zone

Figure 18 shows the cracking around the construction joint between the ceiling cutout and the basement wall for cycles 2-1-N and 2-1-S. Cycle 2-1-N again initiates sliding shear displacements due to plastic crack opening in the joint. A substantial sliding shear displacement occurs then in cycle 2-1-S, with a value of approximately 4 mm. In this cycle, the crack opens on the tension side up to 16 mm. On the compression side, the crack closes incompletely; a plastic deformation remains, which supports the sliding shear failure.

## 4. Analysis

This section aims to roughly analyse the stress and deformation state shortly before sliding shear displacements occur in the clamped part of the walls. Plastic lengthening occurs for the first time in the 2-1-N half-cycle, followed by the first sliding shear displacements in the 2-1-S half-cycle. It is assumed that the main shear force is transferred over the compression zone, because most parts of the reinforcement in the tension zone are already in the plastic range and a significant resistance due to dowel action or inclined lengthening is not left. From a moment–curvature analysis, the compression zone height, x, and the lever arm, z, are available. F_V_ is the maximum shear force in cycle 2-1-N. Sliding shear displacements occurred, for NW 2 and NW 3, when the mean shear stress in the compression zone was τ_x_ ~ 20 MPa or greater. For NW 1, the mean shear stress was 15.2 MPa. No relevant sliding shear displacements occurred. To obtain the friction coefficient, the mean axial stress in the compression zone σ_x_ is calculated. The wall NW 2 with boundary reinforcement resists in terms of amount greater compression stresses (second last column in Table 4). This is due to the greater amount of steel inside the boundary compression zone and its confinement. The friction coefficient in the compression zone is calculated with τ_x_/|σ_x_| or F_V_/|C| and given in the last column of Table 4. There, the concrete and steel resistances are included. Therefore, it can only be a rough global value: sliding shear failure occurred for F_V_/|C| > 0.6. A more detailed analytical analysis can be found in [16].

In Table 5, the deformations around the construction joint of the clamped wall part are analysed. With each “plastic” cycle, the plastic lengthening of the zone around the construction joint proceeds. Already after cycle 2-1-N, the first permanent deformation occurred after unloading. In the tension edge zone (100 mm away from the edge), the lengthening was between 2 mm and 4 mm for walls that fail under sliding shear (see also Figure 13 left and Figure 17 left). By dividing these lengthenings by the measurement length of 200 mm, the tensile strains are obtained, which are greater than 10 mm/m for NW 2 and NW 3. For NW 1, which did not fail through sliding shear, the tensile strain was 8.5 mm/m.

It can be stated that a sliding shear failure only occurs if the shear stress is sufficiently high and a significant plastic flexural deformation is reached. This was the case when τ_x_ > 20 MPa and ε_s_ > 10 mm/m for the investigated walls.

## 5. Conclusions

Shear walls clamped into the basement are very effective for the horizontal stabilisation of buildings against earthquakes. However, the conventional design often considers only the cantilevered part, not the clamped part. This means that the benefit of additional deformation capacity, as well as the risk of premature sliding shear failure in the clamped area, is neglected. This article examines three shear walls that are clamped into the basement box. Particular attention is paid to the analysis of the clamping part.

### 5.1. NW 1: Existing Wall, Slightly Reinforced/Aspect Ratio in the Clamped Part h_w_/l_w_~1

NW 1 is a continuously reinforcement wall with a ratio of 0.44% over the total length. In the early load stages, the wall developed uniform distributed cracks in the cantilever and clamping parts. Later, a large crack opened above the ceiling cutout. Below the ceiling cutout, where the shear stress was high, the crack opening stayed small. No sliding shear failure occurred. The wall failed due to bending. The friction loading in the compression zone was τ_x_/|σ_x_|~0.6 for the maximum load; the strain in the tension zone ε_s_ = 8.5 mm/m.

### 5.2. NW 2: New Construction Wall with Confined Boundaries and a Slightly Reinforced Web/Aspect Ratio in the Clamped Part h_w_/l_w_~1

In NW 2, more cracks occurred compared to NW 1. Especially in the boundary section, secondary cracking occurred due to the higher reinforcement amount. The boundary section has a reinforcement ratio of 2.23%, the web part of 0.44%. In cycle 2-1-N, a flexural crack opened in the construction joint below the ceiling cutout. Under reverse loading, in cycle 2-1-S, the crack closed again, but not completely; NW 2 started to slide. At the maximum load, the friction loading was τ_x_/|σ_x_|~0.75 and the strain in the tension zone ε_s_ = 10.5 mm/m.

### 5.3. NW 3: Existing Wall, Slightly Reinforced/Aspect Ratio in the Clamping Zone = h_w_/l_w_~0.78

NW 3 was 33% longer than NW 1 and NW 2. The reinforcement was continuously distributed with a ratio of 0.39%, nearly the same as in NW 1. As in NW 2, a pronounced crack opening occurred in the construction joint below the ceiling cutout. The opening was mainly on the north side, where a little tension force was introduced from the stilt in the lower support axis. Sliding shear displacements already occurred in cycle 2-1-S, again after the first pronounced lengthening of the tension zone in the half-cycle before. At the maximum load, the friction loading was τ_x_/|σ_x_|~0.78 and the strain in the tension zone ε_s_ = 20 mm/m.

### 5.4. Summary and Outlook

The experiments show that basement-clamped shear walls can fail due to sliding shear in the construction joint below the basement ceiling. A rough estimate can be achieved using the aspect ratio: if the wall is squat enough, h_w_/l_w_ ≤ 1, a sliding shear failure can occur. However, this is a very rough estimate based on only the external geometry. For a closer look, it makes sense to consider the internal deformations and forces of the walls. Under a relevant earthquake loading, it must be assumed that bending cracks open up well into the plastic range (ε_s_ > 10 mm/m). Then, the reinforcement in the tensile zone is clearly deformed into the plastic range and cannot absorb additional sliding shear forces. Further, under reverse loading after plastic crack opening, the cracks do not close completely. This leads to a reduction of the concrete resistance due to aggregate interlock and the shear resistance of the compression reinforcement. Hence, the deceive resistance, the resistance of the compression zone, is reduced. A limit value for the friction coefficient in the compression zone can be specified: sliding shear failure is unlikely if the friction coefficient F_V_/|C| = τ_x_/|σ_x_| ≥ 0.6. This is a global value and includes the resistances of the concrete and the reinforcement in the compression zone and can be easily applied in practice. It is a criterion that includes not only the aspect ratio, as mentioned above, but also the following effects:Amount of reinforcement, which influences the compression resultant C.Distribution of the reinforcement (evenly distributed or concentrated in the edges), which influences the inner lever arm and, as a result, the compression resultant C.Axial force, which influences the compression resultant C.

The compression resultant C can be calculated from the moment–curvature relationship using a commercial cross-section analysis program.

Further research could analyse an additional axial force or the clamping effect of the surrounding ceiling plate.

## Figures and Tables

**Figure 1 materials-17-04111-f001:**
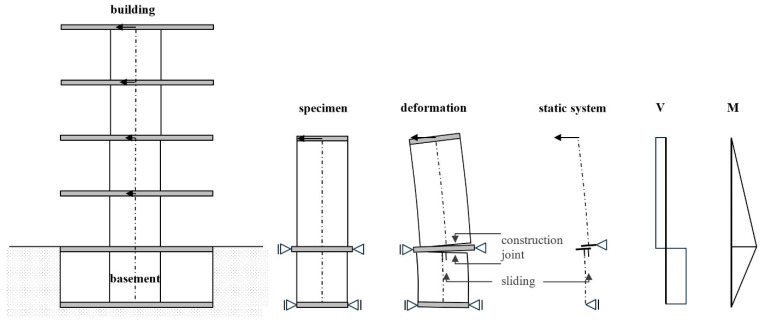
Shear wall clamped in the basement: internal forces, curvatures, and deformation with sliding shear failure [1].

**Figure 2 materials-17-04111-f002:**
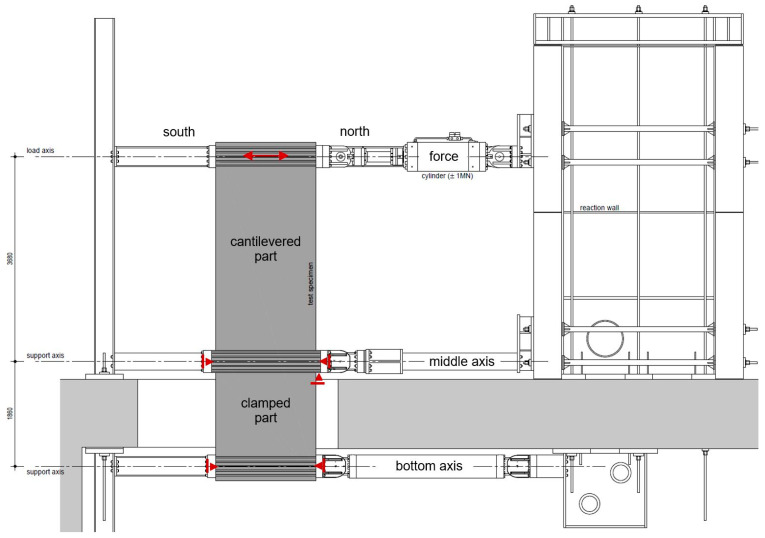
Front view of the experimental set-up [18].

**Figure 3 materials-17-04111-f003:**
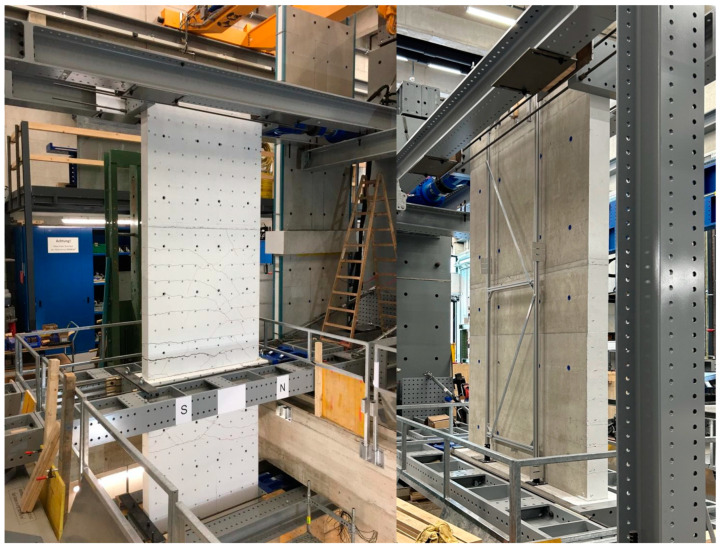
Front side of the test specimen (**left**); back side of the cantilever part of the test specimen with the displacement transducer measuring frame mounted on the ceiling cutout (**right**).

**Figure 4 materials-17-04111-f004:**
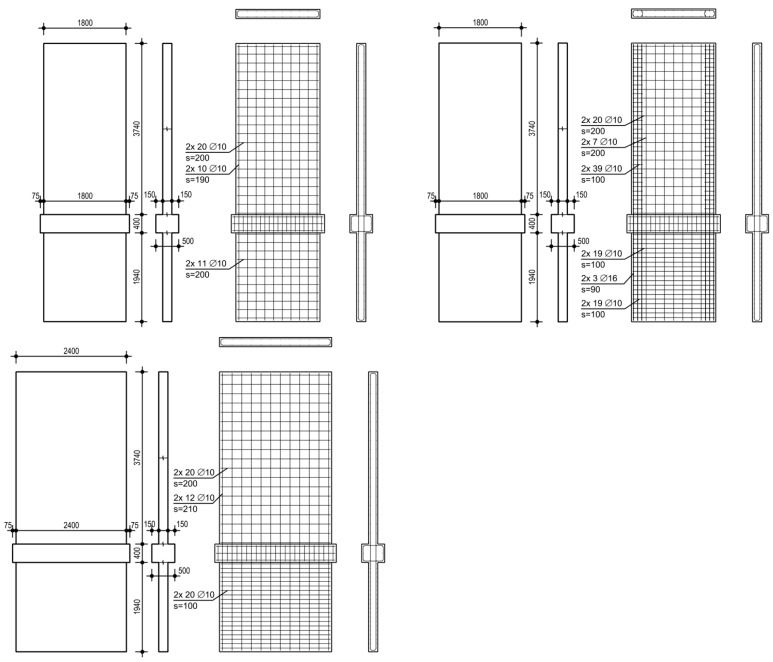
Formwork plan and reinforcement plan of NW 1 (**top left**), NW 2 (**top right**) and NW 3 (**bottom left**); dimensions are in mm.

**Figure 5 materials-17-04111-f005:**
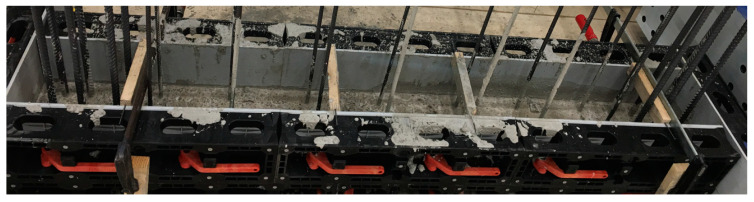
Construction joint between the basement wall and the basement ceiling cutout: complete cross-section (**top**), south side (**bottom left**), and north side (**bottom right**).

**Figure 6 materials-17-04111-f006:**
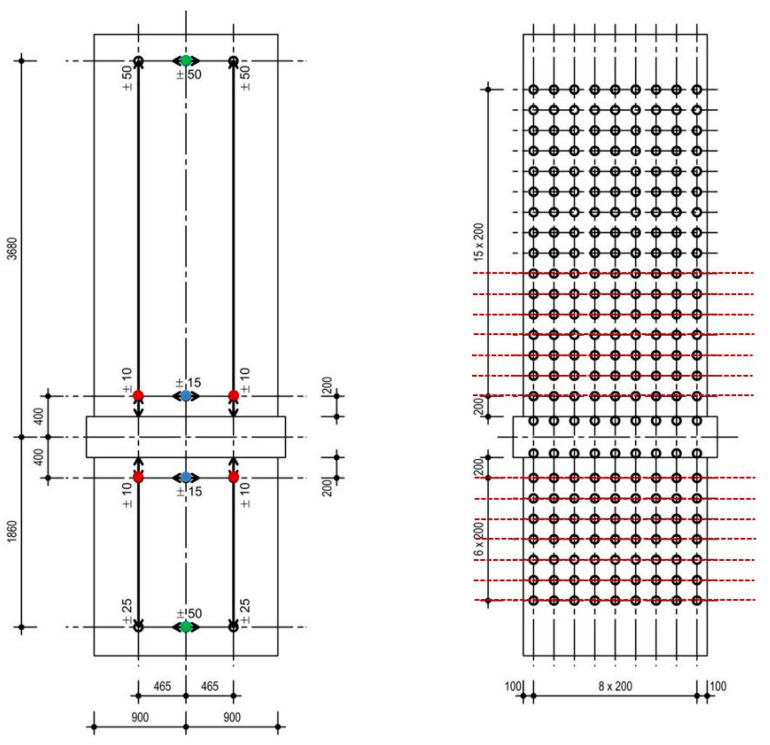
Measurement points of the displacement transducers (**left**): green for the top and bottom displacements, blue for the sliding shear displacements and red for the relative Y-displacements around the construction joints; reflector grid of the optical measurement (**right**); dimensions are in mm; example NW 1.

**Figure 7 materials-17-04111-f007:**
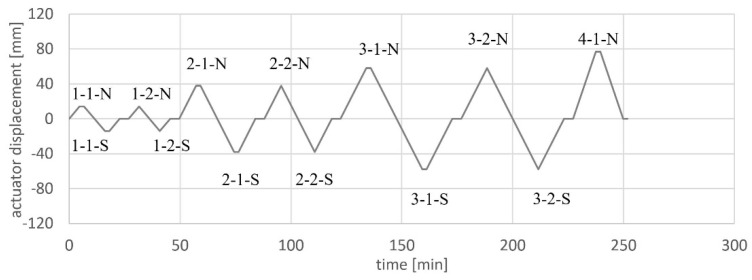
Loading protocol for NW 1.

**Figure 8 materials-17-04111-f008:**
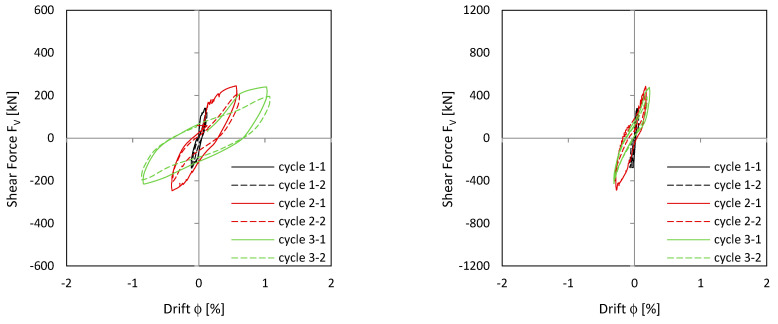
Load–drift curves of NW 1: flex cantilever (**top left**), flex clamping part (**top right**), sliding shear cantilever (**middle left**), sliding shear clamping part (**middle right**), total cantilever (**bottom left**), and total clamping part (**bottom right**).

**Figure 9 materials-17-04111-f009:**
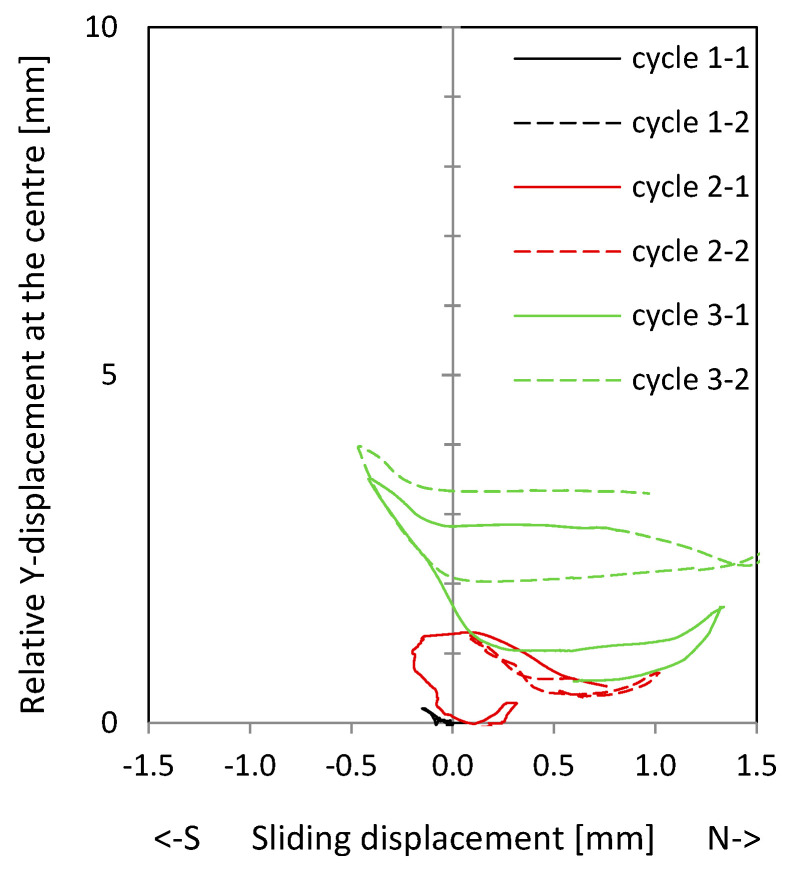
Relative Y-displacement (over a length of 200 mm) to sliding shear displacement curves in the centre of NW 1: above (**top**) and below the ceiling plate (**middle**); permanent relative Y-displacement (over a length of 200 mm) in the centre after the cycle (**bottom**).

**Figure 10 materials-17-04111-f010:**
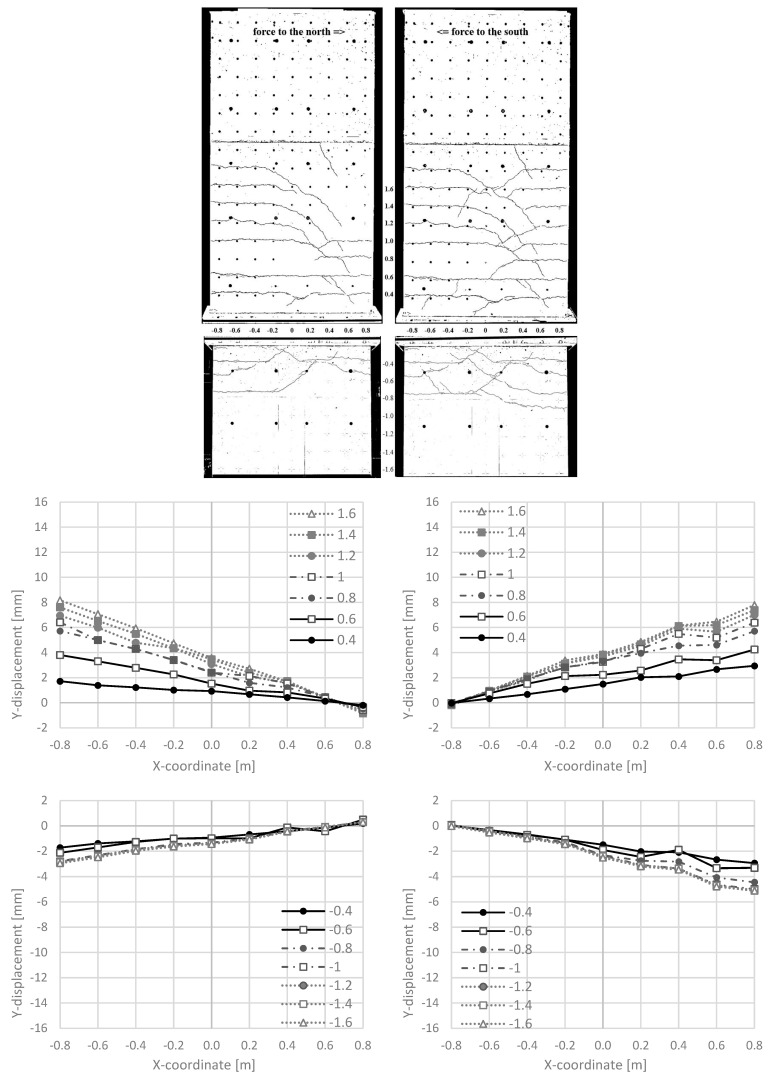
NW 1 in cycle 2-1: to the north (**left**) and to the south (**right**): crack pattern (**top**), Y-displacements along different horizontal sections [m] in the cantilever part (**middle**) and the basement part (**bottom**).

**Figure 11 materials-17-04111-f011:**
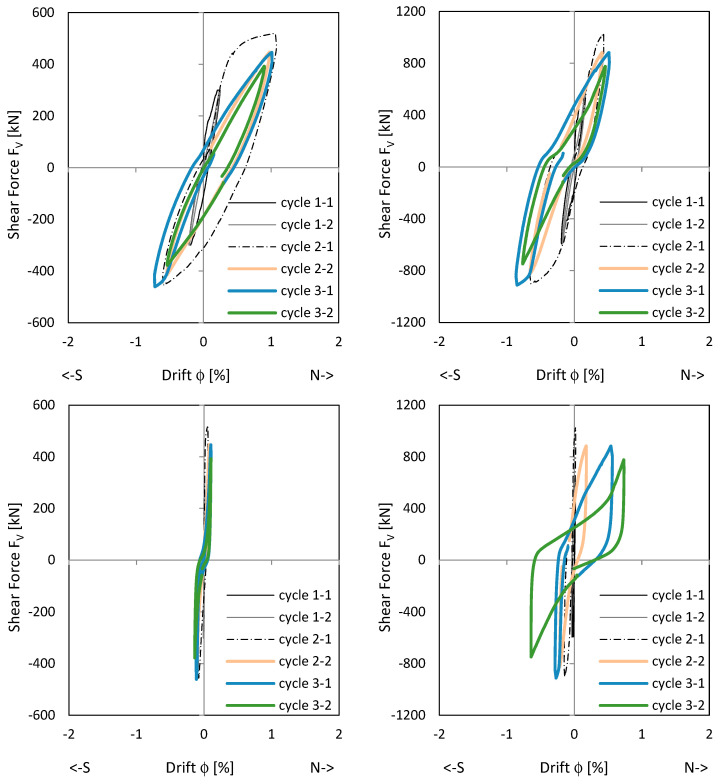
Load–drift curves of NW 2: flex cantilever (**top left**), flex clamping part (**top right**), sliding shear cantilever (**middle left**), sliding shear clamping part (**middle right**), total cantilever (**bottom left**), and total clamping part (**bottom right**).

**Figure 12 materials-17-04111-f012:**
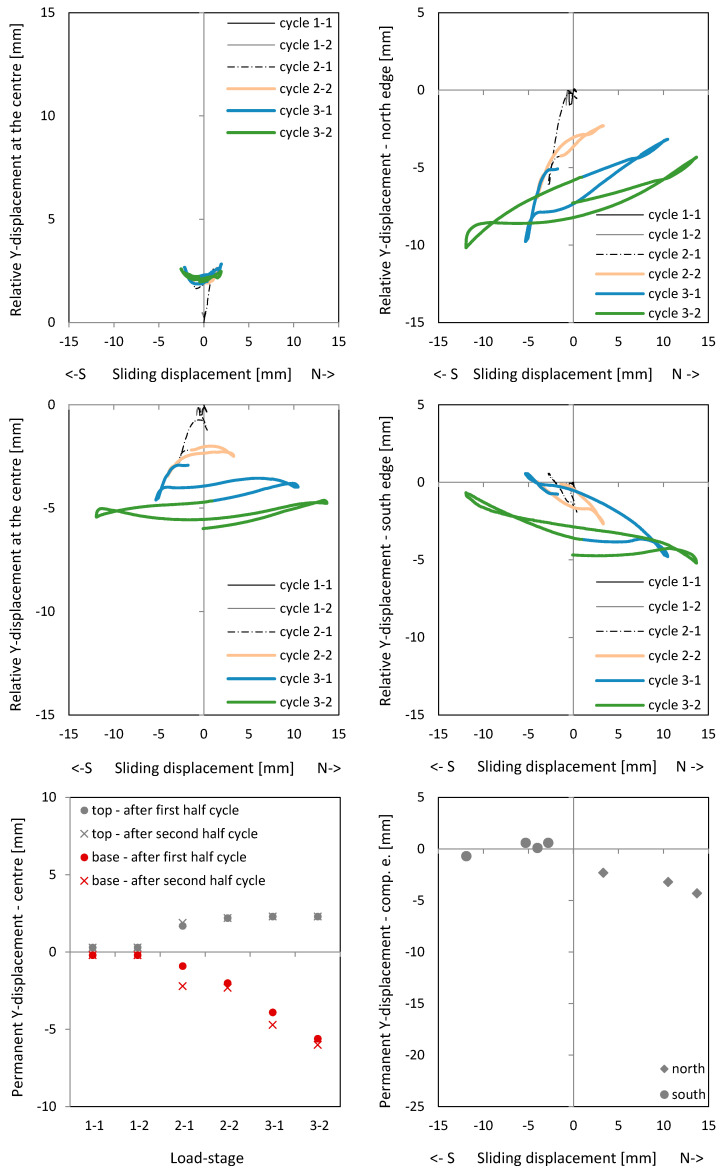
Relative Y-displacement (over a length of 200 mm) to sliding shear displacement curves of NW 2: above (**top left**) and below (**middle left**) the ceiling plate in the centre, at the north edge (**top right**) and at the south edge (**middle right**) of the wall; permanent relative (over a length of 200 mm) Y-displacement after the cycle: in the centre of the wall (**bottom left**) and in the compression zone edge (**bottom right**).

**Figure 13 materials-17-04111-f013:**
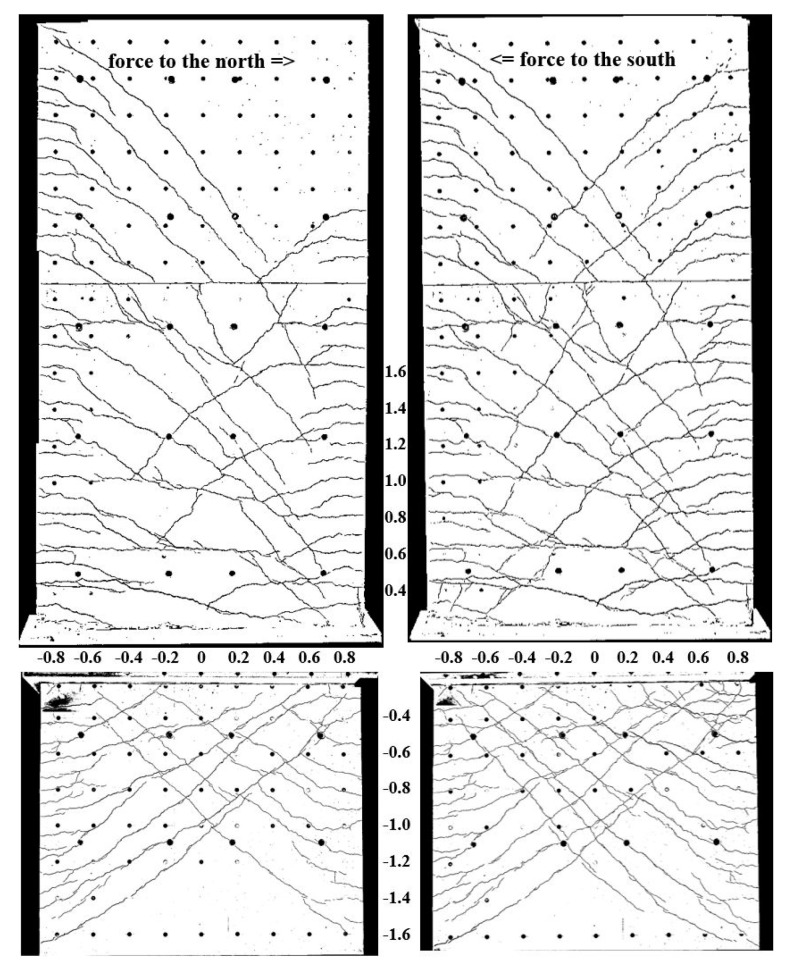
NW 2 in cycle 2-1: to the north (**left**) and to the south (**right**): crack pattern (**top**), Y-displacements along different horizontal sections [m] in the cantilever part (**middle**) and the basement part (**bottom**).

**Figure 14 materials-17-04111-f014:**
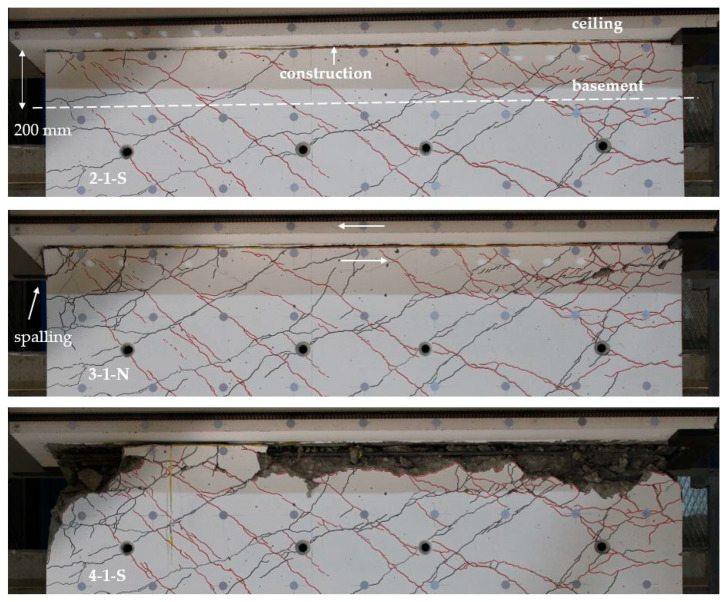
Construction joint between the basement wall and the ceiling cutout of NW 2: cycle 2-1 load to the south (**top**); cycle 3-1 load to the north (**middle**), after cycle 4-1 (**bottom**).

**Figure 15 materials-17-04111-f015:**
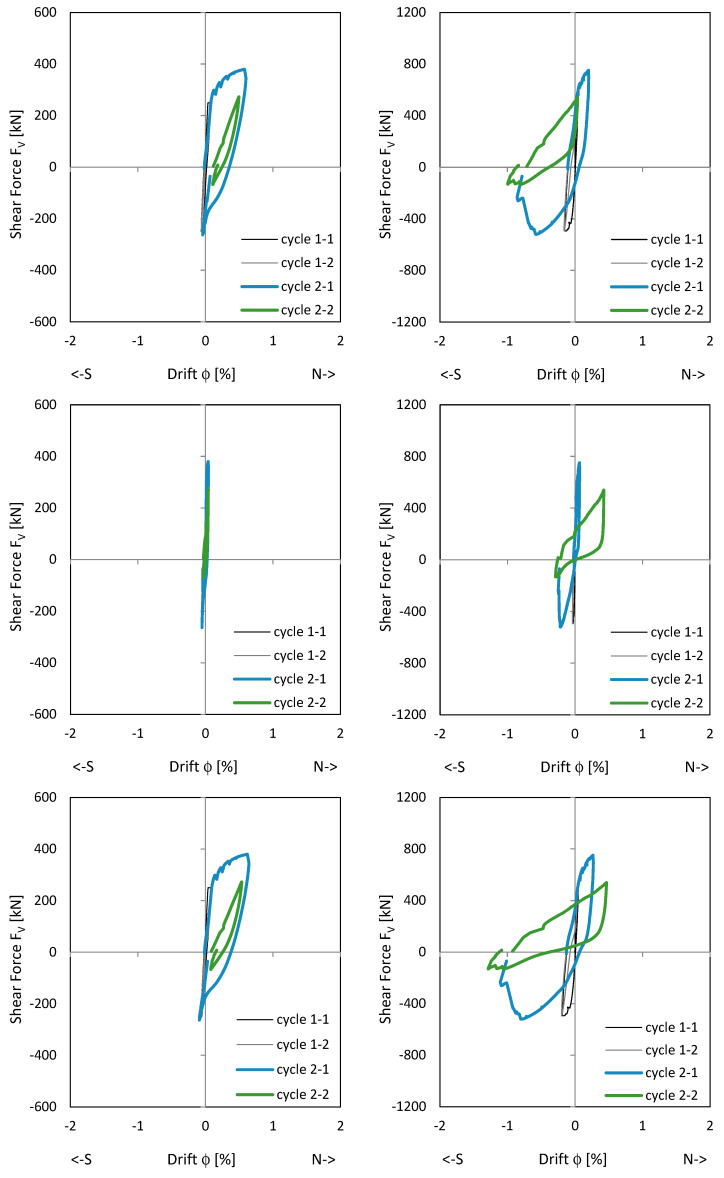
Load–drift curves of NW 3: flex cantilever (**top left**), flex clamping part (**top right**), sliding shear cantilever (**middle left**), sliding shear clamping part (**middle right**), total cantilever (**bottom left**), and total clamping part (**bottom right**).

**Figure 16 materials-17-04111-f016:**
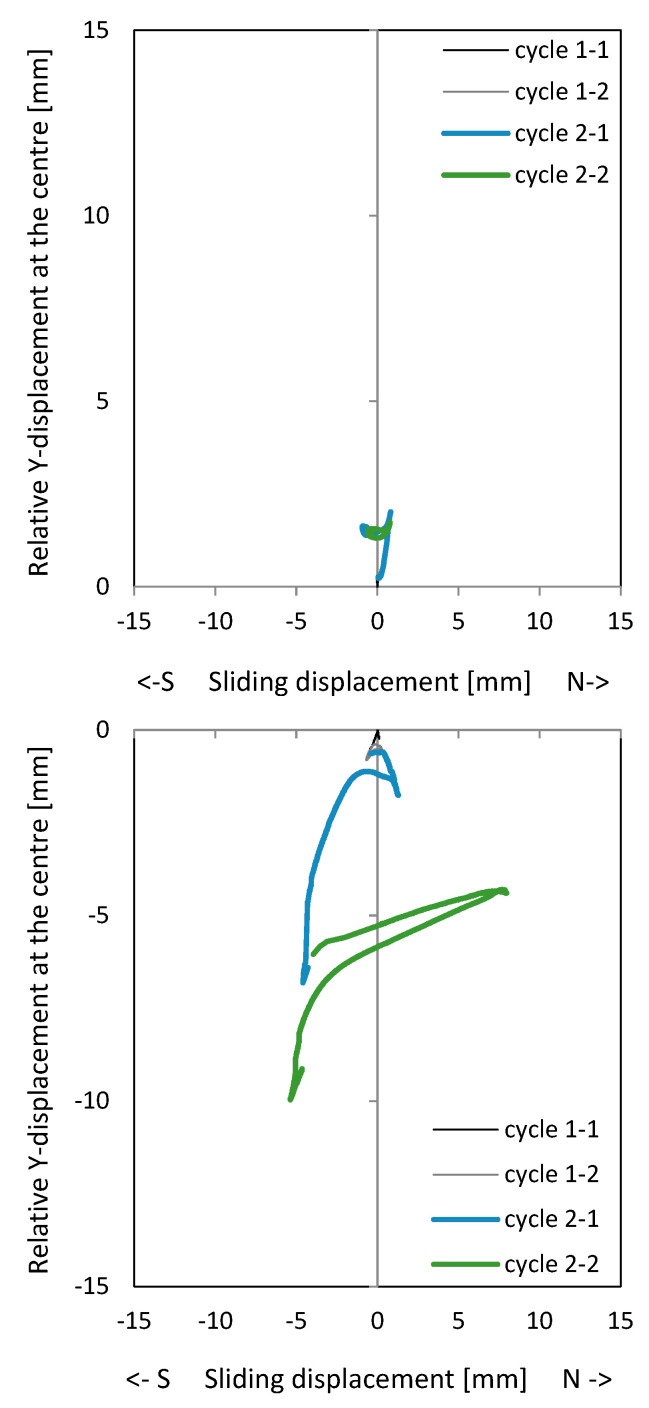
Relative Y-displacement (over a length of 200 mm) to sliding displacement curves in the centre of NW 3: above the ceiling plate (**top**); below the ceiling plate (**middle**); permanent relative Y-displacement (over a length of 200 mm) in the centre after the cycle (**bottom**).

**Figure 17 materials-17-04111-f017:**
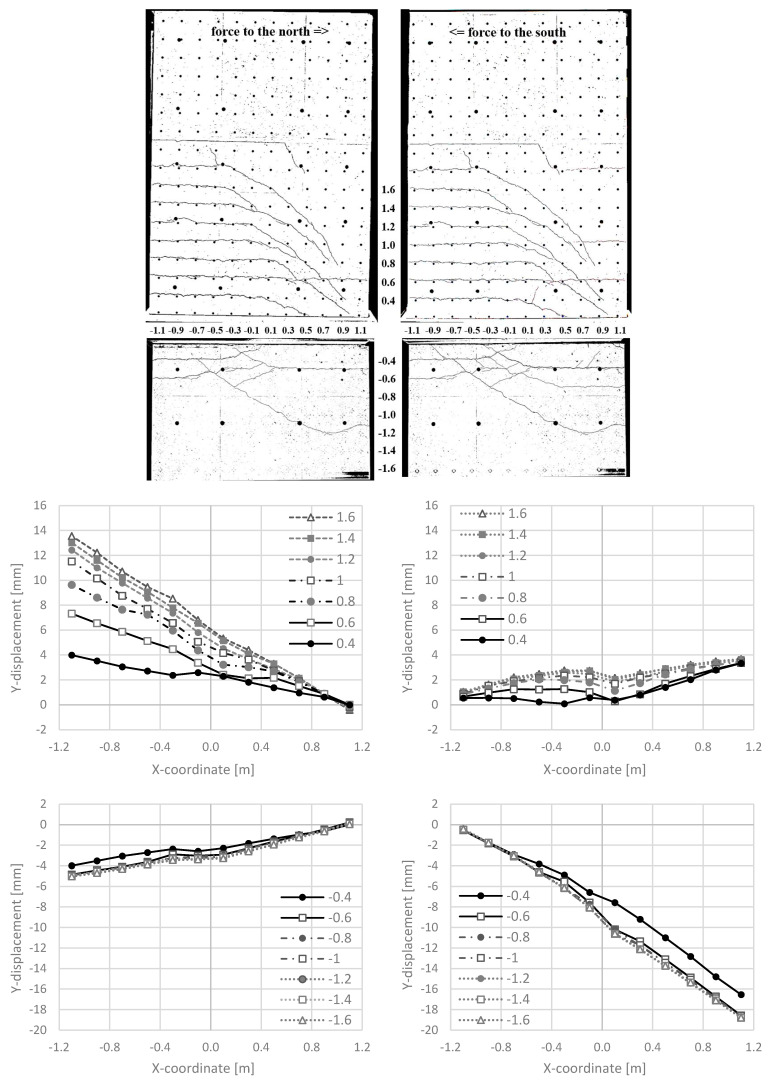
NW 3 in cycle 2-1: to the north (**left**) and to the south (**right**): crack pattern (**top**), Y-displacements along different horizontal sections [m] in the cantilever part (**middle**) and the basement part (**bottom**).

**Figure 18 materials-17-04111-f018:**
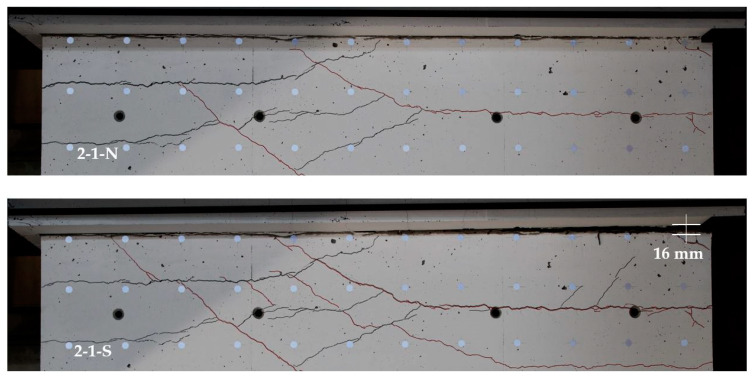
Crack opening of the construction joint below the ceiling cutout of NW 3: at cycle 2-1-N (**top**) and at cycle 2-1-S (**bottom**).

**Table 1 materials-17-04111-t001:** Geometry and reinforcement of the shear walls.

Wall	h_up_	h_base_ = h_w_	t	l_b1/2_	l_web_	ρ_b1/2_	ρ_web_	ρ_V_up_	ρ_V_base_
	[m]	[m]	[m]	[m]	[m]	[%]	[%]	[%]	[%]
1	3.68	1.86	0.20	0	1.80	-	0.44	0.39	0.39
2	3.68	1.86	0.20	0.27	1.26	2.23	0.44	0.39	0.79
4	3.68	1.86	0.20	0	2.40	-	0.39	0.39	0.79

h_up_ = height of the cantilever wall from the middle support axis to the upper load axis; h_base_ = height of the basement wall from the lower to the middle support axis; t = thickness of the wall; l_b_ = length of the boundary zone; l_web_ = length of the web; ρ_b_ = reinforcement ratio in the boundary zone; ρ_web_ = reinforcement ratio in the web; ρ_V_up_ = transverse reinforcement ratio in the cantilever wall; ρ_V_base_ = transverse reinforcement ratio in the basement wall.

**Table 2 materials-17-04111-t002:** Material properties of the concrete at the date of testing.

Wall	f_c_cube_base_	f_c_cube_up_	f_st_base_	f_st_up_
	[MPa]	[MPa]	[MPa]	[MPa]
1	42.2	38.3	3.1	2.7
2	52.6	35.9	4.1	3.2
4	44.1	39.8	3.3	2.9

f_c_cube_base_ = cube compressive strength in the basement wall; f_c_cube_up_ = cube compressive strength in the cantilever wall adjacent to the ceiling cutout; f_st_base_ = splitting tension strength in the basement wall; f_st_up_ = splitting tension strength in the cantilever wall adjacent to the ceiling cutout.

**Table 3 materials-17-04111-t003:** Material properties of the reinforcement.

Ø	E_s_	f_sy_	f_su_	ε_y_	ε_u_
	[GPa]	[MPa]	[MPa]	[mm/m]	[mm/m]
10	202	601	635	3.0	65.3
16	186	606	658	3.3	119

E_s_ = Young’s modulus; f_sy_ = yield strength; f_su_ = tensile strength; ε_y_ = yield strain; ε_u_ = ultimate strain.

**Table 4 materials-17-04111-t004:** Forces, stresses and friction coefficients in the construction joint between the basement wall and the ceiling cutout.

Wall	Cycle	t_w_	x	z	F_V_	Τ = F_V_/(t_w_∙l_w_)	τ_x_ = F_V_/(t_w_∙x)	C = −F_V_∙h_w_/z	σ_x_ = C/(t_w_∙x)	τ_x_/|σ_x_|
		[m]	[m]	[m]	[kN]	[MPa]	[MPa]	[kN]	[MPa]	[-]
NW 1	2-1 N	0.2	0.16	0.98	485	1.3	15.2	−822	−25.7	0.59
NW 2	2-1 N	0.2	0.18	1.24	1135	3.2	31.5	−1519	−42.2	0.75
NW 3	2-1 N	0.2	0.19	1.29	750	2.1	19.7	−965	−25.4	0.78

t_w_ = wall thickness; x = compression zone height; z = lever arm; h_w_ = distance between lower support axis and construction joint below the ceiling cutout = 1.66 m; F_V_ = maximum shear force; C = compression resultant; τ = mean shear stress over the complete cross-section; τ_x_ = mean shear stress over the compression zone; σ_x_ = mean axial stress over the compression zone.

**Table 5 materials-17-04111-t005:** Deformations and strains in the construction joint between the basement wall and the ceiling cutout.

Wall	Cycle	w_s_	ε_s_ = w_s_/0.2 m	Κ = ε_s_/(l_w_-0.1m-x)
		[mm]	[mm/m]	[mrad/m]
NW 1	2-1 N	1.7	8.5	5.5
NW 2	2-1 N	2.1	10.5	6.9
NW 3	2-1 N	4.0	20	9.5

w_s_ = lengthening of the tension side (100 mm away from the edge–outer reflectors from optical measurement) measured over a measurement length of 200 mm around the construction joint; l_w_ = wall length; ε_s_ = tensile strain; κ = curvature.

## Data Availability

Data can be provided on request.

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
