# Peer review of "Sliding Shear Failure of Basement-Clamped Reinforced Concrete Shear Walls"

_materials, 2024, doi:10.3390/ma17164111_

Round 1
Reviewer 1 Report
Comments and Suggestions for Authors
The author investigates the Sliding Shear Failure of Basement-Clamped Reinforced Concrete Shear Walls, which is very interesting. The author conducted an exhaustive experimental study; thus, I think it should be published. However, the writing logic and method of this article should be greatly improved from my point of view.
At first, I do not think the abstract is detailed enough and structurally incomplete. A good abstract is very important and should tell the readers research background, research content, the methodology used in the study and the main conclusion. In your abstract, it is hard for me to find your research conclusions and how did you get it.
Secondly, I think it is logical to write the specimen design first and then the experimental set-up. Besides, it is hard for me to find the research variable in the paper, the research variable is very important and should be easily found in the paper.
Thirdly, it seemed more like a detailed test report for me. I like the test. However, I think more analysis should be presented in this paper. That is what the readers want to see. What happened compared specimen 1, 2 and 3?
Finally, there are three questions presented at the beginning, could you give a clear answer for question 1 and 3? How are the kinematics when sliding shear failure occurs? How is the friction resistance in the compression zone in an already opened construction joint?
Thank you and have a nice day!
Author Response
Please see the attachement.

Reviewer 2 Report
Comments and Suggestions for Authors
General Comments:
This study investigates the sliding shear failure of basement-clamped reinforced concrete shear walls based on three large-scale shear wall tests and proposes a simple estimate for when sliding shear failure can occur. The topic is of relevance to the seismic engineering practice, and the experiments and data analysis are generally well-executed. It is recommended for publication after addressing the following points:
Detailed comments:
1. (Section 2.2, Paragraph 5) Some punctuation marks are in red.
2. (Section 2.2, Paragraph 5) Why is the ultimate strain of Φ16 rebar (119) much higher than that of Φ10 rebar (65.3)?
3. (Section 2.4, Paragraph 1) Please discuss the rationale behind the loading protocol or provide relevant citations, especially for the NW3 that may undergo little cycle. Is this expected in design or an experimental phenomenon?
4. The arrangement of load-drift curves for NW1 (Fig. 8, ‘Total’ at the top) is different from those for NW2 and NW3 (Fig. 11 and 15, ‘Total’ at the bottom).
5. How will the ceiling or floor and its connection type affect the behaviour of the shear wall? As some previous study (JCSR 2023, 204, 107842) found that the flooring system will affect the resistance of the members. In this context, the authors are encouraged to add some discussions on this point.
6. Please discuss the possible reasons for the experimental phenomenon that the NW2’s permanent crack opening or ‘cracks do not close completely’ leading to sliding shear.
7. (Summary and outlook) This study may need more data to draw this conclusion: ‘If the wall is squat enough, hw/lw ≤ 1, a sliding shear failure can occur’.
8. (Summary and outlook) How does the limit FV/C≤0.6 consider the influence of the axial force? And this conclusion seems to be proposed in other publications [16]?
Comments on the Quality of English LanguageThe overall quality of English writing needs to be improved.
Reviewer 3 Report
Comments and Suggestions for Authors
Dear author,
your manuscript brings valuable data. My comments focus on the presentation of results, considering well-established terminology of fracture mechanics and mechanics.
Specific comments:
Missing mix design – cement, type, amount, aggregates, water etc. This has consequences for fracture energy, for example.
The largest grain size of the concrete is 16 mm. → Maximum size of aggregate is 16 mm.
Load-displacement diagrams are normally used. I see no point why to use Load-drift diagrams, the drift is not defined in the article (assuming it is horizontal displacement divided by wall length). Please use Load-displacement in Figure 8, 11, 15.
The term “sliding zone” should be used with caution. More often it refers to “shear zone”.
Figure 8 should be in colors like Figure 11 and 15. I suggest to change x-axis scale to see the changes. Keeping the same scale makes in some sub-figures un-noticeable changes.
Figure 9 – y-axis should be scaled so the crack opening is trackable.
“The vertical displacement is referred to as a crack opening, although several crack openings can occur.” Those are two different things that are incorrect, resulting to things such as permanent crack opening -2 mm (impossible, crack opening could be only positive). Please use the term “vertical displacement”. I’m missing information on crack width at all, at least some indications should be given based on your experiments.
“The shear force is too small to initiate a sliding failure” → The shear force is too small to initiate a shear failure. The title “Sliding Shear Failure of Basement-Clamped Reinforced Concrete Shear Walls” is then incorrect and should be, for example, “Failure of Basement-Clamped Reinforced Concrete Shear Walls under Unconfined Shear”.
Section 4 is highly misleading and should be removed or reworked substantially. You are assuming that tensile part transfers no shear, which is not true due to reinforcement. Another assumption of constant shear stress across compressive part is not valid either since crack opening differs. Make a 2D FEA of your configuration and from that you can describe stress state, crack opening etc. Otherwise, the results are too confusing. Compressive stress should have minus sign and stress MPa units (not MPA).
“It was surprising that the wall with more reinforcement slides rather than the wall with less reinforcement. Perhaps additional walls with a higher reinforcement amount could provide more insights.” What is author’s explanations? Different mode of failure? Statistical scatter? Inconclusive summary statement.
Please release your data as a supplementary material. It is impossible to extract information from the sub-figures. If someone wants to model your experiment, she/he will need detailed data.
Reviewer 4 Report
Comments and Suggestions for Authors
Observations and suggestions for the different sections
Abstract
The abstract could be clearer and more precise. It currently lacks a concise description of the study's main findings and contributions. Consider restructuring the abstract to clearly state the problem, methods, results, and conclusions.
The opening sentence refers to the Chile earthquake of 1985 but does not specify why this event is relevant or how it relates to the study. Providing context or relevance to the current study would enhance understanding.
The abstract mentions that sliding shear failure in the clamping zone of stabilizing walls has received little attention but does not explicitly state why this issue is significant. Emphasize the impact or consequences of this type of failure to justify the study's importance.
The statement "Deformation states around the sliding zone are evaluated" is vague. Specific findings or key results should be highlighted to give readers a sense of the study's contributions. For example, what were the critical deformation states identified, and how do they relate to sliding shear failure?
The abstract ends with a mention of a "simple estimate" for predicting sliding shear failure but does not elaborate on the significance of this estimate. Clarifying how this finding contributes to the field, or its potential applications would strengthen the abstract.
The reference "[1]" is included but not defined within the abstract. If a reference is necessary, ensure it is complete or consider incorporating relevant information directly into the text to maintain the abstract's self-contained nature.
Introduction
The introduction is somewhat disorganized, with a mix of background information, references to previous studies, and specific details about the experimental setup. It would benefit from a clearer structure: start with a broad overview, then narrow down to the specific problem and previous work, and finally introduce the current study's objectives.
While the introduction references numerous studies, the connections between them and their relevance to the current study are not always clear. It's important to succinctly highlight key findings from the literature and how they relate to the present research question.
Some technical details, such as the "tension shift effect," "aspect ratio," and "aggregate interlock resistance," are mentioned without sufficient explanation. This could confuse readers unfamiliar with these concepts. Brief explanations or definitions would help clarify these points.
There are several grammatical errors and awkward phrasings, such as "one finds several experiments on cantilever shear walls, neglecting the clamping effect" and "the deformed specimen in the middle shows." Revising these for clarity and fluency would improve readability.
Experimental set-up and test specimens
The section is dense with information, making it challenging to follow. It would benefit from clearer organization, breaking down the details into more digestible subsections or bullet points where appropriate.
The descriptions of the experimental setup and test specimens contain technical jargon and specific details that may not be immediately clear to all readers. Providing brief explanations or context for terms like "optical measurement reflectors," "displacement transducers," and "tension shift effect" would improve comprehension.
There are inconsistencies in the level of detail provided. For example, NW 1 is described with specifics about reinforcement ratios and aspect ratios, but these details are not consistently provided for NW 2 and NW 3. Ensure that all specimens are described with the same level of detail and clarity.
The measurement techniques (inductive displacement transducers and optical measurement) are described in a way that might be confusing. Clarify how these measurements are used to assess different aspects of the wall's behavior and why multiple measurement techniques were necessary.
The loading protocol description for NW 1 is technical and could be clarified, especially regarding the significance of terms like "0.75Fy," "cycle 1-1 and cycle 1-2," and "δact." Explaining the rationale behind the chosen loading stages and how they relate to real-world conditions would be beneficial.
Results
The section is densely packed with technical details, making it difficult for readers to follow the narrative. Breaking down complex information into simpler, more digestible segments with clear headings and subheadings would improve readability.
There are inconsistencies in the use of terms (e.g., "clamping stiffness," "support points," "basement," "construction joints"). Consistent terminology helps maintain clarity and precision.
Technical terms like "sliding shear failure," "tension shift," and "flexure" are used but not consistently explained or defined. A brief definition or context for these terms would help readers unfamiliar with the subject matter.
The discussion of results sometimes jumps between different concepts (e.g., load-drift behavior, cracking, crack openings) without sufficient transition or explanation of why these observations are significant.
There is a lack of explanation for why certain phenomena occur, such as the asymmetry in load application and its effects. The reasons behind these observations should be clearly stated.
There are several instances where grammar and sentence structure could be improved for better clarity and readability. For example, "The crack openings above and below the ceiling are up to 4 mm." could be rephrased to "Crack openings above and below the ceiling reach up to 4 mm."
Analysis
The section is dense with technical details, making it challenging to follow. Consider breaking down complex sentences and organizing the information more clearly.
Ensure that all terms and abbreviations (e.g., "FV," "tx," "sx") are defined upon first use and consistently used throughout the text.
The references to tables and figures should be clearer and more consistent. For example, refer to specific tables and figures by their numbers (e.g., "Table 1" or "Figure 11") rather than generic descriptions like "Table 1 and 2" or "the figures."
Ensure that all figures and tables mentioned are correctly numbered and captioned. There seems to be a missing reference to "Table 4" and "Table 5," which should be clearly labeled and described in the text.
The analysis should be logically structured, starting with an overview of the findings, followed by a detailed discussion of the results. Currently, it jumps between different aspects (e.g., sliding failure, crack opening, friction coefficient) without a clear progression.
Consider separating the discussion of shear stresses, friction coefficients, and deformation analysis into distinct paragraphs or sub-sections to improve readability.
Provide more context and interpretation of the results. For instance, discuss why certain walls (NW 2 and NW 3) exhibited sliding shear failure and what this implies for structural performance.
Clarify the significance of the friction coefficient values and the criteria for determining sliding shear failure.
Ensure consistent use of units and terms throughout the text. For instance, the term "sx" is used without a clear definition or explanation of its significance.
Verify that the numerical values in the tables are accurate and match the descriptions in the text. For example, check the values of shear stress, axial stress, and friction coefficients.
Conclusion
The section should be more concise and focused. The current format, with subsections for each wall type, is descriptive but lacks a unified conclusion. Consider summarizing the key findings and implications in a cohesive manner.
Clearly distinguish between the description of experimental findings and the general conclusions drawn from the study.
The detailed descriptions of each wall's behavior could be condensed into key takeaways, highlighting the differences and similarities observed. For example, the aspects of reinforcement, crack formation, and sliding shear behavior could be summarized in a few sentences.
Provide a clearer interpretation of the results. What do these findings mean for the design of shear walls in seismic regions? How should the conclusions inform practical engineering practices?
The statement about sliding shear failure criteria (e.g., FV/C = tx/sx ≤ 0.6) should be clearly articulated as a key result of the study and discussed in the context of its practical application.
Ensure consistent use of terminology and symbols. For instance, the use of "tx/sx" should be clearly explained, and its significance should be made explicit.
Verify the accuracy and relevance of numerical values mentioned, such as strain values and friction coefficients, and ensure they are consistently represented.
The language could be more precise and assertive, particularly when stating the conclusions. For example, instead of saying "It can be assumed," state the conclusion more definitively based on the data.
Avoid overly technical jargon that may not be familiar to all readers or ensure that such terms are clearly defined earlier in the paper.
Round 2
Reviewer 1 Report
Comments and Suggestions for Authors
The author has revised the paper according to the reviewer's comments and suggested publication
Author Response
Many thanks again for reading the revised manuscript and many thanks also for the endorsement for publication.
Reviewer 3 Report
Comments and Suggestions for Authors
Dear author,
you have answered several comments which I can accept. However, there are still five comments that need you attention, particularly:
Comment 5: Several sub-figures in Figure 8 are impossible to decode without colors and appropriate scales. Having six lines, black & white, and in low scale is just a mess. You can add colors to represent a cycle, not necessarily sliding displacement. Graphs should be clear what they express at first glance.
Comment 6: Figure 9 – x-axis must be scaled. I see no problem comparing Figures 9, 12, 16 even with different scales – we are talking about range, for example, from -1.5 to 1.5 mm in Figure 9, which is order of magnitude different.
Response 7: “Crack opening” is misleading term in your case. It is defined for a single crack, usually at its mouth, you can consult any literature from fracture mechanics. In elasticity, you have vertical displacement yet no crack. You may argue that tensile strength is relatively low and total strain approximately equals to fracturing strain for higher values. Even “vertical displacement” is incorrect, since you measure relative displacement and not absolute (over the distance of 200 mm). Correct term could be “Relative vertical displacement”. Please, leave defined terms in fracture mechanics and do not redefine them in weird manner.
Response 9: It has been established since 1822 that normal stress is positive in tension. So, put negative stress to Table 4 and put minus sign in your last column. Again, there is no reason to redefine well-established things. (There are some exceptions like stability).
Response 11: “If someone is interested, she/he can contact me, and I provide data.” Please release the data with the article since several graphs are impossible to digitize. I doubt you can provide data after 50 years, data should follow FAIR principles — Findability, Accessibility, Interoperability, and Reusability.
Author Response
Please consider the attached file.

Reviewer 4 Report
Comments and Suggestions for Authors
Upon reviewing the latest version of the manuscript, I can confirm that the author has addressed all the observations and suggestions previously provided.
Author Response

(The authors gave the same response as above.)
